# Influence of Body Weight at the End of the Brooding Period on the Productive Performance in Hyline Brown Laying Hens from 6 to 72 Weeks of Age

**DOI:** 10.3390/ani15091292

**Published:** 2025-04-30

**Authors:** Jian Lu, Qiang Wang, Meng Ma, Yongfeng Li, Wei Guo, Xin Zhang, Xiaodong Yang, Liang Qu

**Affiliations:** 1Jiangsu Institute of Poultry Sciences, Yangzhou 225125, China; lujian1617@163.com (J.L.); yzwangq117@163.com (Q.W.); meng-2005@163.com (M.M.); liyf0120@163.com (Y.L.); guowei1312@outlook.com (W.G.); 2Xianglong Poultry Development Co., Ltd. of Yangzhou, Yangzhou 225261, China; xiangwan2025@163.com (X.Z.); 18205078417@139.com (X.Y.)

**Keywords:** body weight, laying hen, brooding period, growth performance, productive performance

## Abstract

The growth performance of lighter weight pullets at the end of the brooding period could catch up with that of normal weight pullets during the pre-laying and early laying periods, but the productive performance and the proportion of hens laying more than 250 eggs from 18 to 72 weeks of age was lower than that of normal weight hens. These findings indicate that the body weight (BW) by the end of the brooding period can be a good indicator reflecting individual differences among laying hens and may serve as an important phenotypic indicator for evaluating laying performance and early elimination of unqualified laying hens in layer production. Therefore, it is recommended that pullets weighing 25% or more below the normal flock weight at the end of the brooding period should be culled at this time.

## 1. Introduction

Improving production performance during the entire laying cycle can enhance the economic benefits of laying hens, which becomes the focus of the related industry. Previous studies revealed that after culling hens with low laying performance according to the previous individual laying rate (from 21 to 70 wks of age), the mean laying rate of hens aged 70–82 wks was significantly higher than the target laying rate (90.7% to 93.6% vs. 76.1% to 87.0%) [1]. This study also reported that finding and eliminating the hens with poor laying rate or more rest days was a good way to improve the whole laying cycle performance and to extend the laying period, but it is very difficult to select these hens from flocks of raised hens. One way to avoid this problem is to cull laying hens based on their early body weight (BW) during the brooding, rearing, or pre-laying period.

As for laying pullets, the growth, development, and BW in the brooding, rearing, and pre-laying stages serve as pivotal influencing factors for both physical and sexual maturity when the egg-laying cycle starts, potentially directly impacting on the overall laying performance [2]. Therefore, many studies on evaluating BW of laying pullets near the egg-laying onset for its effects on egg quality, sexual maturity, and the whole laying cycle production performance have been conducted. However, most studies have found that compared to hens with lighter BW at the onset of lay, heavier weight hens have higher average egg weight (EW) and cumulative egg mass together with better eggshell quality, but the laying rate in the whole laying cycle did not differ [3,4,5,6]. Therefore, it may be the case that laying pullets’ BW at the egg-laying onset is not a good indicator of individual differences in overall egg production rate and number of resting days, since different rearing management measures during the growing period can improve the BW and flock uniformity at the onset of lay [7,8]. Thus, whether the growth and development status or BW in the brooding period, as an earlier growth stage, can be used as an important phenotypic indicator reflecting individual differences in the whole laying cycle performance has attracted our attention. BW during the brooding period is positively correlated with the development of digestive tract and bone [9,10]. Consequently, at the termination of the brooding period, the hens with greater BW have better physical development and tend to have faster growth rates in the subsequent growth process compared to lower BW hens [11,12]. If the BW is too low at the end of brooding, the pullets may face nutritional deficiencies during the growing period, which will affect subsequent production performance. However, studies specific for the assessment of early weight in affecting later production performance have been rarely performed. Therefore, in-depth studies are required to explore the role of BW at the end of brooding in influencing the overall production performance of laying hens, which may be an important indicator for the elimination of unqualified laying hens in layer production.

The hypothesis tested in this study was that BW at the end of the brooding period may serve as an important phenotypic indicator for evaluating laying performance and early elimination of unqualified laying hens in layer production. This experiment aims to uncover the impacts of BW by the end of the brooding period on Hyline Brown laying hens regarding the egg quality and productive performance, including sexual maturity variables and corresponding CVs, daily egg mass, cumulative egg mass, laying rate, total egg number per hen, CV of individual egg number, and proportion of hens in different egg production ranges from 6 to 72 wks of age.

## 2. Materials and Methods

### 2.1. Study Design, Diets, and Husbandry

Totally, 1200, 1-day-old, Hyline Brown laying pullets sourced from a commercial flock (Xianglong Poultry Development Co., Ltd., Yangzhou, China) were housed in a barn with a controlled environment. Each of the chicks was weighed at the age of 6 wks, with a total of 640 chicks sorted into two groups according to BW: normal (460.75 ± 10.82 g) and light (347.96 ± 6.27 g, 75.52% of normal weight). The chicks in each group were assigned equally to eight replicates containing 40 chicks.

The same corn–soybean meal diets (Table 1), which were prepared in appropriate or excess for the nutrient requirements, were provided ad libitum for all laying hens during the age from 6 to 72 wks (National Research Council, 1994) [13]. The experiment lasted for 66 wks, from July 2022 to November 2023, and was performed at Jiangsu Institute of Poultry Science (Yangzhou, China). At the age of 7 to 17 wks, all chicks were housed in groups of five in enriched cages where each bird was provided with a living space of 475 cm^2^ in a pullet house. From 18 to 72 wks of age, individual cages were employed to offer a living space of 712 cm^2^ per bird in a laying house. Light exposure for 10 h began at 7 to 17 wks of age, which was gradually extended to 16 h at 30 wk. The Animal Care and Use Committee of the Poultry Institute issued an approval for the handling protocols of all animals (SYXK(Su)IACUC 2012-2029).

### 2.2. Sample Acquisition Plus Analytical Determination

BW and Uniformity. At 6, 12, 15, 17–24, and 27 wks of age, the chicks were individually measured for BW. The indirect measure CV was applied to assess BW uniformity [7,8,14]. CV = (standard deviation/average BW) × 100%. BW and BW CV at the first egg were also calculated.

Growth Performance. The feed consumption by replicate was detected every week. The average daily feed intake (ADFI), feed conversion ratio (FCR), and average daily weight gain (ADG) were determined by period and cumulatively.

Body Measurements. At the end of 15 and 20 wks of age, the shank length, shank circumference, and body slope length of all pullets in the two groups were measured by a designated person. The body slope length is the distance between the acromion and the ischial tuberosity. The shank length is the straight-line distance from the upper joint of the shank to the point between the third and fourth toe. The shank circumference is the circumference of the middle of the shank.

Sexual Maturity. The daily eggs of an individual hen were counted. EW and EW CV of the first three eggs in different periods were calculated. The age was recorded when the laying rates of 5%, 50%, and 90% were attained by each replicate.

Productive Performance. Records were kept for all hens concerning the daily egg production and EW, and the weekly feed intake was recorded for replicates. The mean EW, egg mass, FCR, laying rate (hen-day egg production), and feed intake were examined periodically and cumulatively. The proportion of laying hens in different egg production intervals aged between 18 and 72 wks was calculated.

The model for plotting laying rate curves [15] was as follows:*y*(*t*) = *ae*^−*bt*^/[(1 + *e*^−*c*(*t*−*d*)^)]
where *y*(*t*) and *t* refer to the laying rate (%) and age (wks), respectively, and the letters a to d correspond to scale variables that are related to the decreasing rate in laying ability, an inverse indicator for sexual maturity variation, and the average age at sexual maturity.

Egg Quality. A total of 64 focal hens that were initially selected from every treatment group in a random manner (8 hens per replicate) were used for collection of freshly laid eggs at 26, 30, 36, 48, 60, and 72 wks of age. The internal and external features of the eligible eggs were recorded for each examination. The eggs were preserved in a room-temperature environment prior to measurement, and a time interval of less than 24 h was applied from the laying of eggs to measurement.

An FHK egg shape determinator (Fujihira Industry Co., Ltd., Tokyo, Japan) was employed to detect the length and width of eggs, so as to compute the egg shape index. Then, an EQ Reflectometer (Fujihira) was utilized to detect eggshell color at three places (blunt region, equatorial region, and sharp region), of which the mean value was obtained for analyses. The evaluation of eggshell strength was implemented using an EggShell Force Gauge supplied by Robotmation Co., Ltd. (Tokyo, Japan). The measurement of yolk color, EW, albumen height, and Haugh unit was completed with the help of an Egg Multi Tester EMT-5200 (Robotmation). Later, the yolk was separated with the albumen for individual weighing. The eggshell with intact eggshell membrane was weighed. With the inner membranes removed, the blunt, equatorial, and sharp regions were measured for eggshell thickness. Finally, the calculation of ratios of eggshell, albumen, and yolk to EW was performed [16].

Diet Assays. Diet samples were ground, screened using a 40-mesh sieve, frozen immediately, and preserved at −20 °C for subsequent analysis. Then, a Model 1356 adiabatic bomb calorimeter (Parr Instrument Company, Moline, IL, USA) was applied to determine the gross energy. The measurement of phosphorus, crude protein, calcium, and amino acid was conducted in accordance with the procedures formulated by AOAC International (2005) [17].

### 2.3. Statistical Analysis

SPSS v16.0 for Windows (IBM, Armonk, NY, USA) was adopted for all data analyses. Detection of the homogeneity of variance and validation of the normality of the data were executed before analysis. Based on a completely randomized design, data comparisons were implemented through one-tailed *t*-tests. The results are expressed by the format of means ± SEM, and *p* < 0.05 was set as the level of significance, unless otherwise specified.

## 3. Results

### 3.1. BW and Uniformity

The BW of chicks when the brooding period terminated (aged 6 wks old) in the normal weight group (NWG) was significantly heavier than that in the lighter weight group (LWG, 25% lower, *p* < 0.01), but the BW CV of chicks in the NWG was greater than that in the LWG (*p* < 0.05), implying the successful modeling of different BWs of chicks as the brooding period ended.

### 3.2. Growth Performance

The ADG of pullets in the LWG was significantly lower than that in the NWG at ages 7 to 12 wks (*p* < 0.05), but was higher at ages 21 to 24 wks (Table 2; *p* < 0.01). The FCR of chicks in the LWG increased significantly compared with that in the NWG at ages 7 to 12 wks (*p* < 0.05), but decreased at ages 21 to 24 wks (*p* < 0.05). From 7 to 24 wks of age, a significant increase in the ADG of chicks (*p* < 0.01) and a significant decrease in the FCR (*p* < 0.05) were observed in the LWG compared with those in the NWG.

The heavier BW of chicks in the NWG at 6 wks of age compared to LWG birds continued at the age of 12 to 15 wks (*p* < 0.001) and extended until 22 wks (Figure 1; *p* < 0.05), while at ages 23, 24, and 27 wks, BW exhibited no significant difference (*p* > 0.05).

Smaller BW CVs of chicks in the LWG similar to that seen at the end of the brooding period were also observed at 19 (*p* < 0.01), 20 (*p* < 0.01), and 21 (*p* < 0.05) wks of age compared with those in the NWG (Figure 2).

### 3.3. Body Measurement

Both the body slope length and the shank circumference of pullets in the LWG were smaller than those in the NWG at 15 wks of age (Table 3; *p* < 0.001, *p* < 0.05, respectively). But the body slope length, shank length, or shank circumference were not significantly different between the two groups at the age of 20 wks (*p* > 0.05).

### 3.4. Sexual Maturity

In the LWG, the age of hens laying the first egg increased, but its CV declined, compared with those in the NWG (Figure 3; *p* < 0.01, *p* < 0.05, respectively). The differences in BW at first egg and EW of the first three eggs were not statistically significant between the two groups (*p* > 0.05). The age of hens producing 5% eggs in the LWG increased compared with that in the NWG (Table 4; *p* < 0.05), while the age at 50% and 90% egg production exhibited insignificant differences, nor was the interval in days to reach production quotients or the number of eggs produced significantly different between groups (*p* > 0.05).

### 3.5. Productive Performance

The laying hens in the LWG produced a decreased total egg number (*p* < 0.05), along with an increased CV of individual egg numbers (*p* < 0.05) compared with those in the NWG (Figure 4).

The laying rate (*p* < 0.05) and egg mass (*p* < 0.01) of hens aged 18–72 wks in the LWG decreased, while the FCR (*p* < 0.01) increased compared with those in the NWG (Table 5). The mean EW, feed consumption, or mortality presented differences of no significance between the two groups (*p* > 0.05). Figure 5 displays the rate curve fitting.Normal weight hens: y (t) = 112.0654 × e^−0.0047 × t^/[(1 + e^−1.1703 × (t − 23.0569)^)], R^2^ = 0.9893.Lighter weight hens: y (t) = 114.4395 × e^−0.0082 × t^/[(1 + e^−1.8307 × (t − 23.5868)^)], R^2^ = 0.9901.

The hens in the LWG had a decreased laying rate except for laying periods at the ages of 18 to 19 (*p* > 0.05) and 56 to 60 wks (*p* > 0.05), compared with that in the NWG (Figure 6).

The proportion of hens with egg production greater than 300 (*p* < 0.05) or between 251 and 300 (*p* < 0.05) was lower in the LWG than those in the NWG (Figure 7). In contrast, the proportion of hens with egg production less than 150 (*p* < 0.05) or between 151 and 200 (*p* < 0.05) was higher in the LWG compared to those in the NWG.

### 3.6. EW and Egg Quality

There was no significant difference in the EW or the CV of EW in the different laying periods between the two treatments except for EW at the ages of 53 to 56 wks (Table 6; *p* < 0.01). Likewise, the fresh eggs of hens produced at the ages of 26, 30, 36, 48, 60, and 72 wks showed no differences in statistical significance in the external and internal qualities between the two treatments, with the exception of yolk color (*p* < 0.05), at the ages of 30 wks and yolk percentage (*p* < 0.05) at the ages of 48 wks, respectively (Table 7).

## 4. Discussion

In this research, compared with normal weight pullets, the lighter weight pullets with increased ADG and improved FCR only occurred at 21 to 24 wks. Therefore, the inter-group difference in BW of pullets at the end of brooding period persisted until 22 wks of age, and then disappeared by 23 wks of age. Also, the body slope length and the shank circumference of the lighter weight pullets at the end of the brooding period were lower than those of the normal weight pullets at 15 wks of age, but the difference gradually disappeared by the age of 20 wks. This indicated that while individual differences in the BW of pullets on completion of the brooding phase may cause variations of subsequent growth performance for a long time during the growing period. Normal weight pullets at the end of the brooding stage may have better physical development and the organs and tissues may be more mature and have greater ability to digest and absorb nutrients compared with lighter weight pullets, so as to promote subsequent growth [18]. There are very few reports specifically studying the effect of BW by the end of the brooding phase on the growth and development of pre-laying pullets. It was found that grouping by weight during the growth period can increase the BW of small-sized hens during the onset of lay [7,8], coinciding with the present experiment results. Furthermore, through an experiment with different initial BW pullets involving beak trimming and sodium butyrate supplementation, García et al. [19] found that the initial BW (33.9 g for light vs. 37.6 g for heavy) at hatch did not affect the growth performance and pullet uniformity in Lohmann Classic brown pullets at varying time points during the age of hatch and 6 wks. BW, body shape, and flock uniformity are important indicators for evaluating the growth and development of laying hens, and they are also important factors affecting the egg production [2]. The body slope length, shank length, and shank circumference are indicators that reflect the skeletal development of laying hens, and good skeletal development aids laying hens in maintaining good posture and activity level during the brooding period [20,21]. The BW CV reflects the flock uniformity, and good flock uniformity is conducive to improving the growth performance of laying hens. In this experiment, the flock uniformity in lighter weight pullets was better at 19, 20, and 21 wks of age than that of the normal weight pullets. The results indicated that the growth performance of lighter weight pullets at the end of the brooding period could catch up with that of normal weight pullets during the pre-laying and early laying periods. Therefore, pullets’ BW at the egg-laying onset is not a good indicator as the individual differences disappeared during this time. The BW at the end of the brooding period, as an earlier growth stage, is more likely to serve as an important phenotypic indicator, reflecting individual differences.

The ages of hens producing the first egg and 5% eggs of the lighter weight hens were later compared with those of the normal weight pullets in this experiment. The sexual maturity of poultry is a complex physiological process, which is affected by many factors. Among them, early BW exerts an essential effect on the process of poultry’s sexual maturity [22]. Shifting weight gain of broiler breeder pullets from the pubertal period to the prepubertal period advanced sexual maturity [23]. Unlike the findings of this study, an experiment studying breeder hens from the aspects of laying traits plus genetic parameters for BW found that the genetic correlation between BW at 8 wks and the age at first egg was low (–0.17) [24]. This result indicated that early BW could affect laying hens regarding sexual maturity, but this effect gradually weakens with increasing age. In the present study, the age of laying the first egg was delayed by 6.3 days (light, 162.80 d vs. normal, 156.52 d) and the age at 5% egg production was delayed by 10.67 days (light, 151.50 d vs. normal, 140.83 d) in the lighter weight pullets compared with the normal weight pullets. In combination with growth performance data, the BW rapidly increased from 21 to 24 wks of age, and the development of sexual organs started once an appropriate level was reached, thereby accelerating sexual maturity. The lighter weight pullets at the end of the brooding period had 25% lower BW than the normal weight pullets, while the BW did not differ between the two groups at 23 wks of age (154 to 161 d). As a result, lighter weight at the end of the brooding period made laying hens experience postponement of sexual maturity. Moreover, it is interesting to note that the CV of the individual pullet ages at first egg in the LWG declined compared with that of the normal weight pullets. The CV of the BW and age of hens laying the first egg reflects the uniformity in the egg-laying onset, and good uniformity in the egg-laying onset is expected to lead to good egg production performance [2]. Thus, BW at the end of the brooding period can be used as a phenotypic indicator to reflect the individual differences in sexual maturity age and its uniformity.

In this study, at the age of 18–72 wks, the total egg number, egg mass, laying rate, and cumulative egg mass, together with the CV of individual egg number of laying hens in the LWG, decreased compared with those in NWG. However, most research has focused on pullet BW at the egg-laying onset rather than the end of brooding period, to explore its impact on the whole laying cycle production performance. Heavier weight hens at the onset of lay usually have higher average EW and cumulative egg mass, but cumulative egg production in the whole laying cycle was not different [6,25]. For example, insufficiently weighted (27% lower) White Leghorn hens at 16 wks tend to exhibit incomplete body development and insufficient energy reserves, thereby reducing the cumulative egg mass and shortening the peak egg-laying period [26]. The cumulative egg production at the age of 24–90 wks of higher weight hens at 18 wks (average 1.65 kg) is often higher than that of lighter weight hens (average 1.49 kg) [6]. However, the opposite results have also been reported; for example, Lacin et al. [3] found that after assigning Lohmann White hens to three groups according to BW with light (1400 to 1500 g), medium (1500 to 1600 g), and high (>1600 g) groups at 24 wks, the light group exhibited increased cumulative egg production in contrast to the other weight groups at 54 to 84 wks. This may mean that a suitable BW at the egg-laying onset serves as the dominant influencing factor for egg production [27], rather than just considering whether the weight is high or low. According to these results, then, the BW of pullets at the onset of lay mainly affects EW, but has little impact on the laying rate. The lighter weight hens exhibited a significantly higher decline rate of laying ability than that of the normal weight hens according to the rate curve in this research. Furthermore, the lighter weight hens showed a delayed average sexual maturity age (23.06 vs. 23.59 wks of age) according to the rate curve compared with normal weight hens, which was consistent with the data for the ages of producing the first egg and 5% eggs. The delay in the ages of producing the first egg and 5% eggs shortened the laying period, which may be one of the reasons for the lower egg production in the laying period of the lighter weight hens [8]. Therefore, we speculate that the BW by the end of the brooding period is an important factor for laying hens affecting their egg production performance, and the extent of its influence is greater than that of BW at the egg-laying onset.

Good flock uniformity at the start or in the process of the laying period is found to be the major contributor to increasing egg production [28]. Research has shown that a high degree of uniformity when sexual maturity begins is desirable, due to its capability in greater uniformity in egg production onset and egg-laying persistence [2]. Abbas et al. [29] also found that the hens in a higher uniformity group at the start of lay (75 to 80%) maintained a higher hen-day egg production rate at all ages compared with that of hens in a lower uniformity group. In the present study, the flock uniformity at the age of 19, 20, and 21 wks and the uniformity in age laying the first egg for the lighter weight pullets were more consistent than that of normal weight pullets. However, compared with the normal weight hens, the proportion of lighter weight hens laying more than 250 eggs during the age of 18–72 wks was significantly reduced (69.52% vs. 87.38%), while the proportion of hens laying less than 200 eggs was significantly increased (24.97% vs. 3.76%). The CV of individual egg number for laying hens in the LWG increased, which may be one of the reasons for the lower egg production in the laying period of the lighter weight hens. The CV of individual egg number reflects the uniformity in the production performance; a smaller CV indicates that the laying hens lay more evenly and are expected to have good egg production [2]. These results showed that the weight at the end of the brooding period for lighter weight hens was 25% lower than that of normal weight hens, the production performance of those hens was significantly reduced, and good flock uniformity at 19 to 21 wks and good uniformity in the age of the first egg could not eliminate the impact of low weight on production performance.

## 5. Conclusions

In conclusion, our study showed that lighter weight pullets (when the brooding phase ended) had delayed growth together with later sexual maturity in Hyline Brown layer chicks compared with normal weight pullets, but BW can catch up to normal weight pullets by 23 wks of age, while body size traits can catch up by 20 wks of age. In addition, lighter weight hens had worse production performance, including individual egg number, mean egg mass, laying rate, and the CV of individual egg numbers at age 18–72 wks compared with normal weight hens. Compared with normal weight hens, the proportion of lighter weight hens laying more than 250 eggs during the age of 18–72 wks was significantly reduced (69.52% vs. 87.38%), while the proportion of hens laying less than 200 eggs was significantly increased (24.97% vs. 3.76%). These findings suggest that BW by the end of the brooding period can be a good indicator reflecting individual differences among laying hens and may serve as an important phenotypic indicator for evaluating laying performance and early elimination of unqualified laying hens in layer production. Therefore, it is recommended that pullets weighing 25% or more below the normal flock weight at the end of the brooding period should be culled at this time.

## Figures and Tables

**Figure 1 animals-15-01292-f001:**
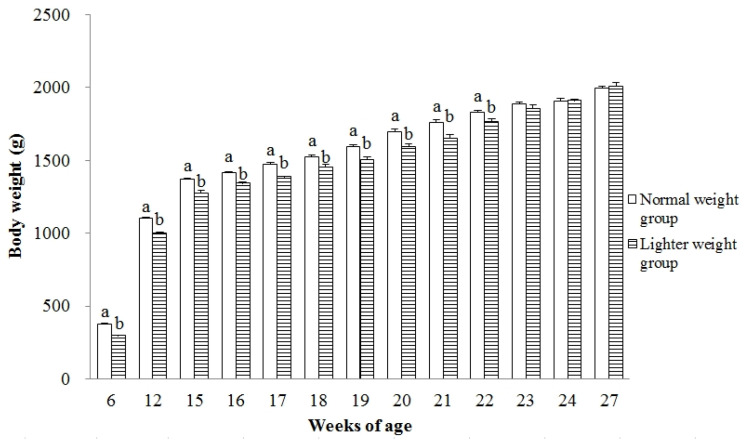
The BW (g) during sexual maturation of hens with different BWs on completion of the brooding phase. Values are means of eight replicates per dietary treatment. Columns with different superscripts (a, b) at the same age are significantly different at *p* < 0.05.

**Figure 2 animals-15-01292-f002:**
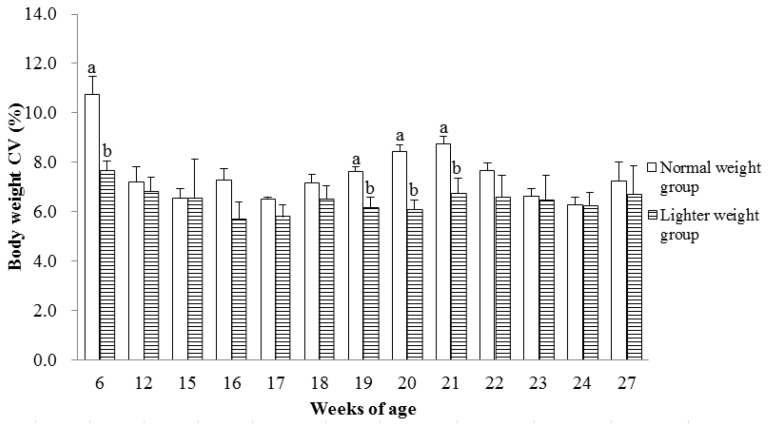
The BW CV (%) during sexual maturation of hens with different BWs when the brooding phase ended. Values are means of eight replicates per dietary treatment. Columns with different superscripts (a, b) at the same age are significantly different at *p* < 0.05.

**Figure 3 animals-15-01292-f003:**
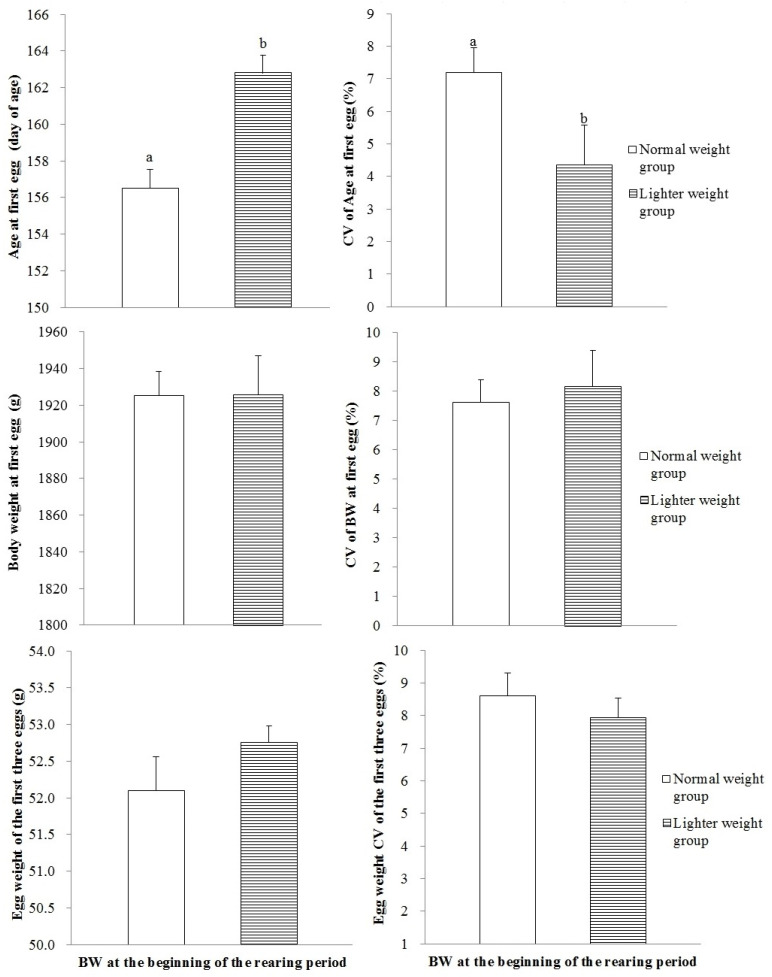
The age of laying the first egg, BW at the time of laying the first egg, and the first three eggs’ weight for hens with different BWs as the brooding period terminated. Values are means of eight replicates per dietary treatment. Columns with different superscripts (a, b) are significantly different at *p* < 0.05.

**Figure 4 animals-15-01292-f004:**
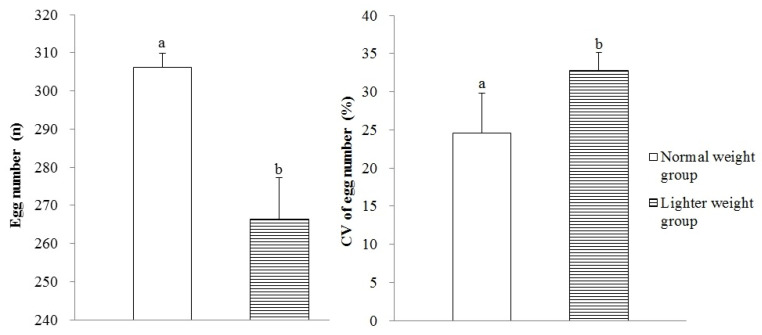
The number of eggs plus CV of individual egg number during the age of 18–72 wks for hens with different BWs on completion of the brooding phase. Values are means of eight replicates per dietary treatment. Columns with different superscripts (a, b) are significantly different at *p* < 0.05.

**Figure 5 animals-15-01292-f005:**
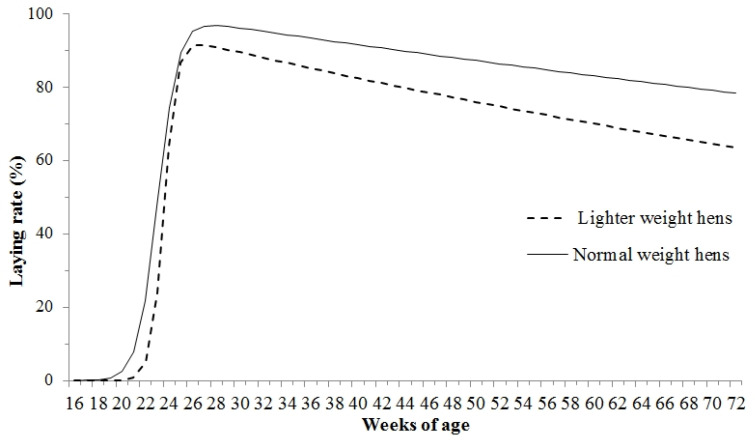
The laying rate curves for hens with different BWs at the age of 18–72 wks at the end of the brooding period.

**Figure 6 animals-15-01292-f006:**
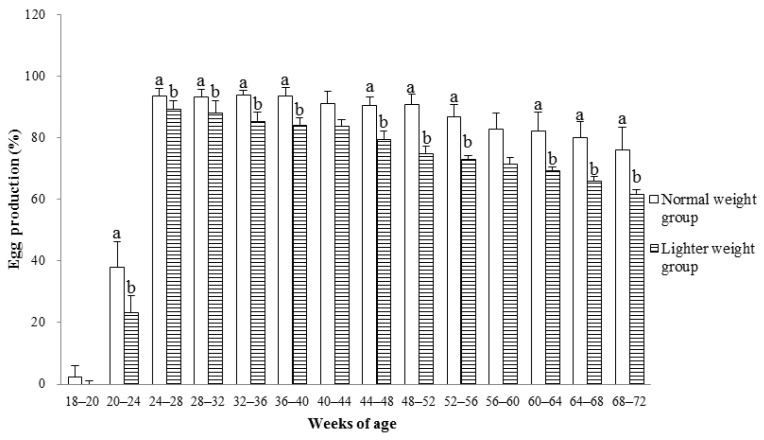
The egg production in different laying periods for hens with different BWs at the termination of the brooding period. Values are means of eight replicates per dietary treatment. Columns with different superscripts (a, b) at the same period are significantly different at *p* < 0.05.

**Figure 7 animals-15-01292-f007:**
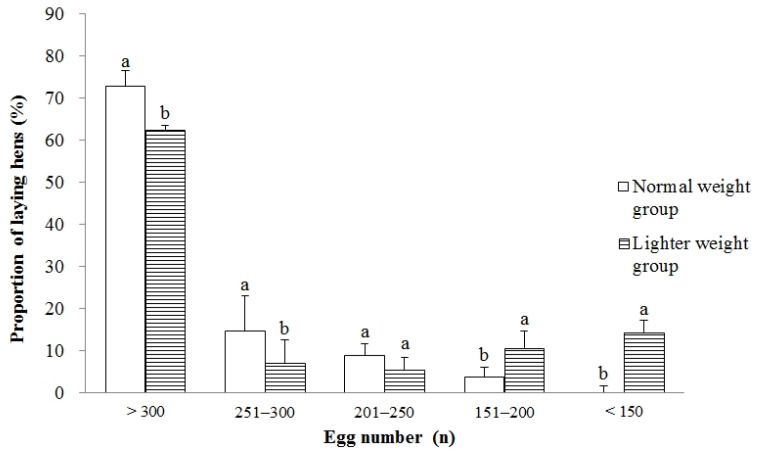
The percentage of hens in different egg production ranges during the age of 18–72 wks for hens with different BWs at the end of the brooding period. Values are means of eight replicates per dietary treatment. Columns with different superscripts (a, b) at the same range are significantly different at *p* < 0.05.

**Table 1 animals-15-01292-t001:** Ingredients and nutrient content of the experimental diet offered during the age of 6–72 wks ^1^.

	6–12 Wks	13–17 Wks	18–20 Wks	21–72 Wks
Ingredient (%)				
Corn	71.11	75.20	68.00	65.50
Soybean meal	25.67	20.80	24.00	23.50
Limestone	1.40	2.00	—	3.00
Calcareous granule	—	0.20	6.00	6.00
Zeolite powder	0.105	0.105	0.23	0.23
Dicalcium phosphate	0.595	0.595	0.105	0.105
Monocalcium phosphate	0.25	0.25	0.595	0.595
Sodium chloride	0.12	0.10	0.25	0.25
50% Choline chloride	0.15	0.15	0.12	0.12
DL-methionine	0.10	0.10	0.20	0.20
Premix of vitamins and trace minerals ^2^	0.50	0.50	0.50	0.50
Nutrient content (calculated)				
AME_n_ (kcal/kg)	2850	2850	2800	2700
Crude protein (%)	17.50	15.50	16.50	16.00
Total amino acid (%)				
Lysine	0.94	0.82	0.84	0.82
Methionine	0.45	0.41	0.47	0.47
Methionine + Cystine	0.73	0.67	0.74	0.73
L-tryptophan	0.20	0.18	0.19	0.19
Threonine	0.66	0.59	0.62	0.61
Calcium (%)	0.80	0.80	2.60	3.80
Total phosphorus (%)	0.50	0.50	0.50	0.50
Nonphytate phosphorus (%)	0.27	0.27	0.27	0.27
Nutrient content (measured)				
DM (%)	91.90	92.17	91.61	92.08
Gross energy (kcal/kg)	4017	4032	3944	3892
Crude protein (%)	17.75	15.61	16.59	16.04
Total amino acid (%)				
Lysine	0.89	0.78	0.81	0.79
Methionine	0.43	0.38	0.48	0.45
Met + Cys	0.74	0.69	0.75	0.70
L-tryptophan	0.19	0.19	0.20	0.18
Threonine	0.64	0.58	0.59	0.59
Calcium (%)	0.84	0.87	2.68	3.78
Total phosphorus (%)	0.53	0.53	0.51	0.54

^1^ Values obtained are presented on an air-dry basis. ^2^ Premix contained (diet in kg): selenium, 0.3 mg; vitamin A, 7715 IU; copper, 8 mg; vitamin D_3_, 2755 IU; zinc, 80 mg; vitamin E, 8.8 IU; biotin, 0.20 mg; cobalamin, 20 μg; pyridoxine, 3.25 mg; riboflavin, 2.21 mg; menadione, 2.2 mg; nicotinic acid, 19.8 mg; folic acid, 0.28 mg; thiamine, 0.65 mg; pantothenic acid, 3.51 mg; manganese, 65 mg; iron, 60 mg; and iodine, 1.0 mg.

**Table 2 animals-15-01292-t002:** The mortality (%), ADG (g/bird per day), ADFI (g/bird per day), and FCR (g/g) during the age of 7–24 wks of the pullets with different BWs at the end of the brooding period ^1^.

Items	Normal Weight Hens	Lighter Weight Hens	*p*-Value
7–12 wks			
ADG	17.37 ± 0.080 ^a^	16.80 ± 0.178 ^b^	0.011
ADFI	60.09 ± 0.104	60.16 ± 0.148	0.653
FCR	3.46 ± 0.019 ^b^	3.59 ± 0.037 ^a^	0.010
13–15 wks			
ADG	12.55 ± 0.286	13.00 ± 0.744	0.532
ADFI	80.92 ± 0.376 ^b^	83.10 ± 0.767 ^a^	0.018
FCR	6.45 ± 0.147	6.46 ± 0.392	0.982
16–17 wks			
ADG	7.63 ± 0.906	6.10 ± 1.300	0.387
ADFI	70.71 ± 0.572	71.81 ± 1.197	0.423
FCR	9.62 ± 1.159	12.71 ± 2.635	0.300
18–20 wks			
ADG	10.07 ± 1.656	11.70 ± 0.300	0.503
ADFI	83.21 ± 1.455	84.80 ± 1.064	0.452
FCR	8.52 ± 1.135	7.28 ± 0.355	0.466
21–24 wks			
ADG	7.65 ± 0.591 ^b^	11.33 ± 0.464 ^a^	0.002
ADFI	100.50 ± 1.093	102.78 ± 0.904	0.177
FCR	13.63 ± 1.197 ^a^	9.12 ± 0.320 ^b^	0.012
7–20 wks			
ADG	13.45 ± 0.189	13.23 ± 0.225	0.469
ADFI	70.99 ± 0.307	72.02 ± 0.427	0.079
FCR	5.29 ± 0.052	5.45 ± 0.080	0.118
7–24 wks			
ADG	12.17 ± 0.128 ^b^	12.80 ± 0.091 ^a^	0.007
ADFI	77.55 ± 0.472	78.86 ± 0.325	0.076
FCR	6.38 ± 0.056 ^a^	6.16 ± 0.041 ^b^	0.020
Mortality (%)	0.63 ± 0.555	2.50 ± 3.191	0.259

^a,b^ Means in the same row that are not marked with common superscripts have significant differences (*p* < 0.05). ^1^ Values stand for the average of eight replicates from each dietary treatment.

**Table 3 animals-15-01292-t003:** Body measurement of pullets with different BWs at the end of the brooding period ^1^.

Items	Normal Weight Hens	Lighter Weight Hens	*p*-Value
15 wk			
Body slope length (cm)	21.43 ± 0.084 ^a^	20.56 ± 0.216 ^b^	<0.001
Shank length (cm)	8.57 ± 0.025	8.53 ± 0.045	0.379
Shank circumference (cm)	3.64 ± 0.012 ^a^	3.60 ± 0.011 ^b^	0.007
20 wk			
Body slope length (cm)	23.38 ± 0.106	23.05 ± 0.130	0.050
Shank length (cm)	8.63 ± 0.030	8.60 ± 0.039	0.525
Shank circumference (cm)	3.82 ± 0.013	3.81 ± 0.016	0.916

^a,b^ Means not marked by common superscripts in the same row are significantly different (*p* < 0.05). ^1^ Values stand for the average of eight replicates from each dietary treatment.

**Table 4 animals-15-01292-t004:** The sexual maturity variables of Hyline Brown laying hens with different BWs at the end of the brooding period ^1^.

Items	Normal Weight Hens	Lighter Weight Hens	*p*-Value
Age at 5% egg production (day of age)	140.83 ± 2.880 ^b^	151.50 ± 2.217 ^a^	0.029
Age at 50% egg production (day of age)	160.17 ± 1.195	163.00 ± 0.913	0.126
Age at 90% egg production (day of age)	171.67 ± 2.044	177.50 ± 3.926	0.185
5% to 50% egg production interval (day)	19.33 ± 3.313	11.50 ± 1.658	0.072
50% to 90% egg production interval (day)	11.50 ± 2.078	14.50 ± 4.031	0.486
5% to 90% egg production interval (day)	30.83 ± 4.400	26.00 ± 5.642	0.514
5% to 50% egg production egg number (*n*)	4.30 ± 0.766	2.43 ± 0.354	0.097
50% to 90% egg production egg number (*n*)	8.72 ± 1.615	11.30 ± 3.471	0.470
5% to 90% egg production egg number (*n*)	13.00 ± 1.936	13.70 ± 3.621	0.857

^a,b^ Means in a row with no common superscripts marked show significant differences (*p* < 0.05). ^1^ Values represent the average of eight replicate sets for each dietary treatment.

**Table 5 animals-15-01292-t005:** The productive performance for hens with different BWs at the age of 18–72 wks at the end of the brooding period ^1^.

Items	Normal Weight Hens	Lighter Weight Hens	*p*-Value
Laying rate (%)	80.25 ± 0.992 ^a^	71.04 ± 2.906 ^b^	0.029
Egg weight (g)	61.37 ± 0.173	61.08 ± 0.482	0.310
Egg mass (g/day)	50.33 ± 0.675 ^a^	43.33 ± 1.463 ^b^	0.001
Feed consumption (g/hen per day)	111.89 ± 0.733	113.62 ± 0.650	0.256
Feed conversion ratio (kg of feed/kg of eggs)	2.21 ± 0.032 ^b^	2.63 ± 0.081 ^a^	0.001
Mortality (%)	2.52 ± 1.218	4.20 ± 1.668	0.432

^a,b^ Means in the same row without common superscripts exhibit differences of significance (*p* < 0.05). ^1^ Values refer to the average of eight replicate sets for each dietary treatment.

**Table 6 animals-15-01292-t006:** The EW (g) and EW CV (%) of hens with different BWs at the end of the brooding period ^1^.

Items	Normal Weight Hens	Lighter Weight Hens	*p*-Value
EW (g)			
21–24 wk	54.85 ± 0.250	54.22 ± 0.241	0.125
25–28 wk	59.52 ± 0.187	58.81 ± 0.491	0.248
29–32 wk	61.83 ± 0.184	61.71 ± 0.309	0.730
33–36 wk	62.90 ± 0.314	62.15 ± 0.629	0.274
37–40 wk	63.75 ± 0.271	63.35 ± 0.648	0.530
41–44 wk	64.04 ± 0.418	63.01 ± 0.646	0.194
45–48 wk	62.51 ± 0.394	61.86 ± 0.826	0.453
49–52 wk	61.89 ± 0.701	61.43 ± 0.554	0.654
53–56 wk	60.77 ± 0.295 ^a^	59.09 ± 0.276 ^b^	0.004
57–60 wk	60.29 ± 0.212	59.79 ± 0.810	0.490
61–64 wk	60.40 ± 0.374	60.69 ± 0.413	0.626
65–68 wk	61.39 ± 0.349	61.85 ± 0.689	0.522
69–72 wk	61.59 ± 0.586	60.88 ± 1.443	0.613
19–72 wk	60.06 ± 0.176	59.69 ± 0.361	0.337
EW CV (%)			
21–24 wk	9.69 ± 0.674	8.47 ± 0.595	0.244
25–28 wk	7.78 ± 0.445	8.41 ± 0.269	0.327
29–32 wk	7.80 ± 1.043	8.08 ± 0.917	0.855
33–36 wk	8.69 ± 0.641	7.97 ± 1.003	0.539
37–40 wk	7.62 ± 0.545	7.29 ± 0.272	0.601
41–44 wk	7.15 ± 0.744	7.40 ± 0.831	0.828
45–48 wk	7.32 ± 0.545	8.49 ± 0.172	0.088
49–52 wk	7.59 ± 0.498	7.42 ± 0.260	0.806
53–56 wk	7.77 ± 0.498	8.36 ± 0.700	0.499
57–60 wk	8.26 ± 0.481	8.83 ± 0.614	0.482
61–64 wk	8.12 ± 0.411	8.26 ± 0.375	0.816
65–68 wk	7.96 ± 0.620	8.88 ± 0.872	0.401
69–72 wk	7.68 ± 0.797	7.52 ± 0.844	0.899
19–72 wk	9.28 ± 0.386	9.21 ± 0.343	0.909

^a,b^ Means that are not indicated by common superscripts in the same row are significantly different (*p* < 0.05). ^1^ Values stand for average levels of eight replicates from different dietary treatments.

**Table 7 animals-15-01292-t007:** The egg quality of hens with different BWs at the end of the brooding period ^1^.

Items	Normal Weight Hens	Lighter Weight Hens	*p*-Value
26 wks of age			
Eggshell color	22.83 ± 0.679	22.73 ± 0.987	0.926
Egg shape index	1.27 ± 0.008	1.27 ± 0.007	0.512
Average shell thickness (mm)	0.41 ± 0.005	0.41 ± 0.009	0.575
Blunt end thickness (mm)	0.40 ± 0.008	0.40 ± 0.011	0.976
Equator thickness (mm)	0.40 ± 0.006	0.41 ± 0.008	0.434
Acute end thickness (mm)	0.42 ± 0.004	0.42 ± 0.010	0.598
Eggshell strength (Kg/cm^2^)	4.83 ± 0.105	4.77 ± 0.318	0.873
Yolk color	8.08 ± 0.260	7.88 ± 0.350	0.632
Albumen height (mm)	7.94 ± 0.471	7.37 ± 0.470	0.434
Haugh unit	89.07 ± 2.997	86.32 ± 2.670	0.543
Eggshell percentage (%)	10.47 ± 0.140	10.38 ± 0.212	0.713
Yolk percentage (%)	23.41 ± 0.404	23.33 ± 0.412	0.886
Albumen percentage (%)	66.11 ± 0.474	66.06 ± 0.517	0.944
30 wks of age			
Eggshell color	22.88 ± 0.752	24.06 ± 0.428	0.190
Egg shape index	1.26 ± 0.007	1.28 ± 0.013	0.379
Average shell thickness (mm)	0.40 ± 0.007	0.41 ± 0.004	0.098
Blunt end thickness (mm)	0.40 ± 0.008	0.41 ± 0.007	0.734
Equator thickness (mm)	0.40 ± 0.008	0.42 ± 0.006	0.084
Acute end thickness (mm)	0.40 ± 0.008	0.42 ± 0.005	0.064
Eggshell strength (Kg/cm^2^)	4.59 ± 0.099	4.91 ± 0.306	0.348
Yolk color	5.50 ± 0.261 ^a^	4.75 ± 0.164 ^b^	0.026
Albumen height (mm)	8.63 ± 0.264	7.86 ± 0.439	0.128
Haugh unit	91.98 ± 1.240	87.54 ± 2.675	0.109
Eggshell percentage (%)	9.72 ± 0.173	10.15 ± 0.119	0.052
Yolk percentage (%)	25.04 ± 0.585	25.77 ± 0.350	0.387
Albumen percentage (%)	65.24 ± 0.609	64.08 ± 0.429	0.198
36 wks of age			
Eggshell color	25.15 ± 1.235	24.08 ± 1.302	0.569
Egg shape index	1.27 ± 0.004	1.29 ± 0.014	0.169
Average shell thickness (mm)	0.39 ± 0.005	0.39 ± 0.008	0.615
Blunt end thickness (mm)	0.38 ± 0.007	0.39 ± 0.008	0.472
Equator thickness (mm)	0.39 ± 0.005	0.40 ± 0.008	0.850
Acute end thickness (mm)	0.39 ± 0.006	0.39 ± 0.010	0.973
Eggshell strength (Kg/cm^2^)	4.10 ± 0.250	4.09 ± 0.160	0.976
Yolk color	7.92 ± 0.358	6.88 ± 0.479	0.093
Albumen height (mm)	7.66 ± 0.642	7.16 ± 0.493	0.581
Haugh unit	85.37 ± 4.245	83.56 ± 2.931	0.756
Eggshell percentage (%)	9.63 ± 0.140	9.68 ± 0.140	0.816
Yolk percentage (%)	26.47 ± 0.376	26.23 ± 0.740	0.775
Albumen percentage (%)	63.89 ± 0.410	64.09 ± 0.698	0.799
48 wks of age			
Eggshell color	26.75 ± 1.336	24.56 ± 0.905	0.241
Egg shape index	1.29 ± 0.008	1.30 ± 0.014	0.612
Average shell thickness (mm)	0.39 ± 0.014	0.37 ± 0.011	0.339
Blunt end thickness (mm)	0.39 ± 0.014	0.36 ± 0.009	0.231
Equator thickness (mm)	0.40 ± 0.011	0.38 ± 0.008	0.137
Acute end thickness (mm)	0.39 ± 0.017	0.38 ± 0.017	0.705
Eggshell strength (Kg/cm2)	3.83 ± 0.284	3.55 ± 0.212	0.480
Yolk color	5.36 ± 0.244	5.29 ± 0.286	0.841
Albumen height (mm)	6.96 ± 0.375	7.31 ± 0.220	0.450
Haugh unit	82.48 ± 2.339	85.32 ± 1.389	0.334
Eggshell percentage (%)	9.66 ± 0.329	9.19 ± 0.266	0.326
Yolk percentage (%)	26.09 ± 0.385 ^b^	27.56 ± 0.293 ^a^	0.013
Albumen percentage (%)	64.25 ± 0.662	63.25 ± 0.302	0.187
60 wks of age			
Eggshell color	25.43 ± 0.710	26.19 ± 1.804	0.703
Egg shape index	1.28 ± 0.009	1.26 ± 0.017	0.335
Average shell thickness (mm)	0.37 ± 0.007	0.39 ± 0.009	0.129
Blunt end thickness (mm)	0.36 ± 0.009	0.38 ± 0.010	0.144
Equator thickness (mm)	0.37 ± 0.010	0.39 ± 0.010	0.219
Acute end thickness (mm)	0.38 ± 0.006	0.39 ± 0.010	0.170
Eggshell strength (Kg/cm^2^)	3.67 ± 0.203	3.76 ± 0.313	0.788
Yolk color	5.64 ± 0.338	5.75 ± 0.366	0.824
Albumen height (mm)	6.78 ± 0.382	6.23 ± 0.418	0.354
Haugh unit	81.72 ± 2.589	77.92 ± 2.818	0.346
Eggshell percentage (%)	9.24 ± 0.225	9.90 ± 0.189	0.051
Yolk percentage (%)	27.13 ± 0.528	27.04 ± 0.845	0.918
Albumen percentage (%)	63.63 ± 0.673	63.06 ± 0.848	0.607
72 wks of age			
Eggshell color	24.33 ± 0.860	28.64 ± 2.030	0.080
Egg shape index	1.29 ± 0.011	1.31 ± 0.017	0.473
Average shell thickness (mm)	0.38 ± 0.016	0.38 ± 0.010	0.797
Blunt end thickness (mm)	0.38 ± 0.016	0.37 ± 0.008	0.505
Equator thickness (mm)	0.39 ± 0.016	0.39 ± 0.011	0.986
Acute end thickness (mm)	0.38 ± 0.019	0.37 ± 0.013	0.909
Eggshell strength (Kg/cm^2^)	3.65 ± 0.249	3.72 ± 0.223	0.845
Yolk color	6.08 ± 0.336	6.00 ± 0.378	0.873
Albumen height (mm)	6.84 ± 0.319	6.98 ± 0.255	0.767
Haugh unit	81.07 ± 2.084	82.81 ± 1.796	0.564
Eggshell percentage (%)	9.63 ± 0.428	9.66 ± 0.262	0.960
Yolk percentage (%)	25.48 ± 0.713	25.73 ± 0.501	0.798
Albumen percentage (%)	64.89 ± 0.618	64.61 ± 0.557	0.755

^a,b^ Means in a row not marked with common superscripts exhibit significant differences (*p* < 0.05). ^1^ Values represent average levels of eight replicate sets for each dietary treatment.

## Data Availability

No new data were created.

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
