# Peer review of "Influence of Body Weight at the End of the Brooding Period on the Productive Performance in Hyline Brown Laying Hens from 6 to 72 Weeks of Age"

_animals, 2025, doi:10.3390/ani15091292_

Round 1

Reviewer 1 Report

Comments and Suggestions for Authors

Strength 

Authors investigated the influence of body weight (BW) upon completion of the brooding period on Hyline Brown laying hens from 6 to 72 weeks of age 3 regarding productive performance. Authors show results and describe how BW by the end of brooding period can better reflect individual differences among laying hens and may serve as an important phenotypic indicator for evaluating laying performance and early elimination of unqualified laying hens in layer production. Overall, the manuscript is well written, and the data and findings in this paper are vital to poultry science.

Weakness

N/A

Minor comments and suggestions

  1. Abbreviations such as BW (Line 12) and CV (Line 39) should first be written in full together with the appropriate abbreviation then abbreviated throughout the manuscript.

Author Response

Strength 

Authors investigated the influence of body weight (BW) upon completion of the brooding period on Hyline Brown laying hens from 6 to 72 weeks of age 3 regarding productive performance. Authors show results and describe how BW by the end of brooding period can better reflect individual differences among laying hens and may serve as an important phenotypic indicator for evaluating laying performance and early elimination of unqualified laying hens in layer production. Overall, the manuscript is well written, and the data and findings in this paper are vital to poultry science.

Thanks for your approval.

Weakness

N/A

Thanks for your approval.

Minor comments and suggestions

  1. Abbreviations such as BW (Line 12) and CV (Line 39) should first be written in full together with the appropriate abbreviation then abbreviated throughout the manuscript.

-Done as requested.

Line 12. body weight (BW)

Line 39. coefficient of variation (CV)

Reviewer 2 Report

Comments and Suggestions for Authors
  1. Lines 143-145: Please provide a detailed explanation of how the following parameters were measured:
  • the shank length
  • shank circumference
  • body slope length
  1. Table 2, Pages 6 and 7: Are the authors confident about the accuracy of the FCR data? The data seems quite scattered.
  2. Line 256: The text related to Figure 4 explanations should be separated from the main manuscript text:
    "Columns with different superscripts (a, b) are significantly different at P < 0.05."
  3. Line 320: The word "may" is repeated twice.
  4. Lines 300-341: The discussion on body weight results lacks specific reasoning or hypotheses for the observed results. Please rewrite this section.

Author Response

Lines 143-145: Please provide a detailed explanation of how the following parameters were measured:

the shank length, shank circumference, body slope length

-Done as requested.

At the end of 15 and 20 wks of age, the shank length, shank circumference and body slope length of all pullets in the two groups were measured by designated person. The body slope length is the distance between the acromion and the ischial tuberosity. The shank length is the straight-line distance from the upper joint of the shank to the point between the third and fourth toe. The shank circumference is the circumference of the middle of the shank.

Table 2, Pages 6 and 7: Are the authors confident about the accuracy of the FCR data? The data seems quite scattered.

We have rechecked the original data. The data is accurate. However, the data varied greatly between repetitions, which may be due to the frequent weighing, mainly occurs at 16, 17, 18 weeks of age

Line 256: The text related to Figure 4 explanations should be separated from the main manuscript text:

"Columns with different superscripts (a, b) are significantly different at P < 0.05."

-Done as requested.

Line 320: The word "may" is repeated twice.

-Done as requested.

The word "may" was deleted.

Lines 300-341: The discussion on body weight results lacks specific reasoning or hypotheses for the observed results. Please rewrite this section.

-Done as requested.

In the present research, compared with normal weight pullets, the lighter weight pullets with increased ADG and improved FCR only occurred at 21 to 24 wks. Therefore, the inter-group difference in BW of pullets at the end of brooding period persisted until 22 wks of age, then disappeared by 23 wks of age. Also, the body slope length and the shank circumference of the lighter weight pullets at the end of the brooding period were lower than those of the normal weight pullets at 15 wks of age, but the difference gradually disappeared by the age of 20 wks. This indicated that while individual differences in the BW of pullets on completion of brooding phase may cause variations of subsequent growth performance for a long time during the growing period. Normal weight pullets at the end of brooding stage may have better physical development and the organs and tissues may be more mature and have greater ability to digest and absorb nutrients, so as to promote subsequent growth [18]. There are very few reports specifically studying the effect of BW by the end of brooding phase on the growth and development of pre-laying pullets. It was found that grouping by weight during the growth period can increase the BW of small-sized hens during the onset of lay [7-8], coinciding with the present experiment results. Besides, through an experiment with different initial BW pullets involving beak trimming and sodium-butyrate supplementation, García et al. [19] found that the initial BW (33.9 g for light vs. 37.6 g for heavy) at hatch did not affect the growth performance and pullet uniformity at varying time points during the age of hatch and 6 wks in Lohmann Classic brown pullets. BW, body shape and flock uniformity are important indicators for evaluating the growth and development of laying hens, and they are also important factors affecting the egg production [2]. The body slope length, shank length and shank circumference are indicators that reflect the skeletal development of laying hens, and good skeletal development aids laying hens in maintaining a good posture and activity level during the brooding period [20-21]. The BW CV reflects the flock uniformity, and good flock uniformity is conducive to improving the growth performance of laying hens. In this experiment, the flock uniformity of lighter weight pullets was better at 19, 20, and 21 wks of age than that of the normal weight pullets. It was indicated that the growth performance of lighter weight pullets at the end of the brooding period could catch up with that of normal weight pullets during the pre-laying and early laying periods. Therefore, pullets' BW at the egg-laying onset is not a good indicator as the individual differences disappeared in this time. The BW at the end of the brooding period, as an earlier growth stage, is more likely to serve as an important phenotypic indicator, reflecting individual differences.

Reviewer 3 Report

Comments and Suggestions for Authors

The study has focused on body weight of pullet and its effects on productive performance. However, it is a basic topic especially for genetic companies. Also, this subject has already been worked by many researchers. 

Acoording to the measured parameters, the results are expected, and any innovation has been observed. I mean that it could be better to investigate the mechanism the difference for performance between two groups or what could be the solutions for minimaze of these negative status?

The article does not contain any new information as of this form, and unfortunately ot have potential to contribute to the journal or poultry industry.

Author Response

Comments and Suggestions for Authors

The study has focused on body weight of pullet and its effects on productive performance. However, it is a basic topic especially for genetic companies. Also, this subject has already been worked by many researchers. Acoording to the measured parameters, the results are expected, and any innovation has been observed. I mean that it could be better to investigate the mechanism the difference for performance between two groups or what could be the solutions for minimaze of these negative status? The article does not contain any new information as of this form, and unfortunately to have potential to contribute to the journal or poultry industry.

It is a pity that this study did not get your affirmation.

The purpose of this experiment is to solve the problems encountered in the laying hen industry. We think it is a meaningful study that can guide the breeding of layer hens. So far, there are very few reports specifically studying the effect of BW by the end of brooding phase on the growth and development of pre-laying pullets and the laying rate of hens during the age of 18-72 wks.

From the surface, this experiment may not have too much novelty. However, this experiment provides a lot of detailed data for the laying hen industry, including the growth and development rules of pullet with different BW at 6 wks of age, the rules of changes in group uniformity, the rules of changes in egg production rate, the coefficient of variation of the onset traits, the coefficient of variation of individual egg production rate, and the proportion of layer chickens with different egg production rates. The results of the study show that, BW by the end of brooding period can better reflect individual differences among laying hens and may serve as an important phenotypic indicator for evaluating laying performance and early elimination of unqualified laying hens in layer production. Therefore, it is recommended that pullets weighing 25% or more below the normal flock weight at the end of the brooding period should be culled at this time. The results have important guiding significance for guiding the production of laying hens

Submission of a paper is also a learning process for the authors. We can learn the opinions of reviewers to improve ourselves. We will further study the mechanism of the performance difference between the two groups or possible solutions to minimize these negative states.

Thanks!

Reviewer 4 Report

Comments and Suggestions for Authors

This research study was conducted to evaluate the effects of post-hatching live weight difference on egg production performance and it was revealed that chicks with low live weight had lower performance during the production period.

A simple summary was written in an easy and understandable way.
The summary is very long and full of unnecessary information and takes a little longer to understand. Therefore, it needs to be reorganized in a shorter and more understandable way
In line 34 35, 37, 39, 41, 43……………….. it should be P<0.011 not P=
In line 39 What is CV short for? Why is CV important? CV difference is shown in Figure 2, but the values ​​for 19,20,22 weeks are given just above it. I did not understand CV is a very necessary yield.
Shouldn't it be (P>0.05) in line 45? Because there is no statistical difference.
CV = (standard deviation/average BW) × 100%. BW and BW CV at first egg were also calculated. And how does the CV value affect egg production characteristics or why is it important? There is no explanation in the article?

Table 2 OK
In line 196-202-203-211-218-226-234-237-250-257-258-270-278-79-292  should be P<
Figure 1 OK for an easy-to-understand article
Figure 2 What importance does the BV CV value have for the article that you gave? Is it really necessary?

Table 3 OK but 20 wk Body slop length P value P<0.050 should be checked, there is a significant difference, there is no lettering.

Table 4 OK
CV values ​​are given in Figures 3 and 4 but what is the importance of the CV value is not mentioned in the article. Since the necessary parameter should be explained, if not, it should be removed.

Table 5 OK
Figure 5 and 6 OK
Why is the CV value for egg weights important in Table 6? What does it mean? Is it necessary for the article?
Table 7 OK.
In line 256. Columns with different superscripts (a, b) are significantly different at P < 0.05 should be added above figure
There is no statement in the discussion about the CV of egg counts, but in the conclusion section it is stated with a sentence that the CV of egg counts was worse in the LWG group. What should readers of this article understand about the CV value or what it means should be stated in more detail in the discussion.

Author Response

Comments and Suggestions for Authors

This research study was conducted to evaluate the effects of post-hatching live weight difference on egg production performance and it was revealed that chicks with low live weight had lower performance during the production period.

A simple summary was written in an easy and understandable way.

The summary is very long and full of unnecessary information and takes a little longer to understand. Therefore, it needs to be reorganized in a shorter and more understandable way

-Done as requested.

Simple Summary: The growth performance of lighter weight pullets at the end of the brooding period could catch up with that of normal weight pullets during the pre-laying and early laying periods, but the productive performance and the proportion of hens laying more than 250 eggs from 18 to 72 wks of age was lower than that of normal weight hens. It was indicated that body weight (BW) by the end of brooding period can better reflect individual differences among laying hens and may serve as an important phenotypic indicator for evaluating laying performance and early elimination of unqualified laying hens in layer production. Therefore, it is recommended that pullets weighing 25% or more below the normal flock weight at the end of the brooding period should be culled at this time.

In line 34 35, 37, 39, 41, 43……………….. it should be P<0.011 not P=

-Done as requested.

We have checked and revised throughout the manuscript.

In line 39 What is CV short for? Why is CV important? CV difference is shown in Figure 2, but the values ​​for 19,20,22 weeks are given just above it. I did not understand CV is a very necessary yield.

-Be: coefficient of variation

The BW CV reflects the flock uniformity, and good flock uniformity is conducive to improving the growth performance of laying hens. The CV of the BW and age of hens laying the first egg reflects the uniformity of the egg-laying onset, and a good uniformity of the egg-laying onset is expected to have better egg production performance. The CV of individual egg number of reflects the uniformity of the production performance, a smaller CV indicates that the laying hens lay more evenly and is expected to have a better egg production.

In previous studies, our team found that there was significant individual difference in egg production performance of laying hens, especially when the hens were subjected to different treatments such as environment, nutrition and management. Therefore, in this experiment, individual data were recorded for phenotypic traits of laying hens, such as body weight, laying characteristics, egg production characteristics and so on, in order to analyze the individual differences of laying hens with different BW at 6 wks of age.

Shouldn't it be (P>0.05) in line 45? Because there is no statistical difference.

-Be revised.

CV = (standard deviation/average BW) × 100%. BW and BW CV at first egg were also calculated. And how does the CV value affect egg production characteristics or why is it important? There is no explanation in the article?

-Be revised. Related explanations are supplemented in the discussion section.

The BW CV reflects the flock uniformity, and good flock uniformity is conducive to improving the growth performance of laying hens. The CV of the BW and age of hens laying the first egg reflects the uniformity of the egg-laying onset, and a good uniformity of the egg-laying onset is expected to have better egg production performance. The CV of individual egg number of reflects the uniformity of the production performance, a smaller CV indicates that the laying hens lay more evenly and is expected to have a better egg production.

In previous studies, our team found that there was significant individual difference in egg production performance of laying hens, especially when the hens were subjected to different treatments such as environment, nutrition and management. Therefore, in this experiment, individual data were recorded for phenotypic traits of laying hens, such as body weight, laying characteristics, egg production characteristics and so on, in order to analyze the individual differences of laying hens with different BW at 6 wks of age. 

Table 2 OK

In line 196-202-203-211-218-226-234-237-250-257-258-270-278-79-292  should be P<

-Done as requested.

We have checked and revised throughout the manuscript.

Figure 1 OK for an easy-to-understand article

Figure 2 What importance does the BV CV value have for the article that you gave? Is it really necessary?

Related explanations are supplemented in the discussion section. We believe that it is an important indicator. The BW CV (Coefficient of variation) reflects the flock uniformity, and good flock uniformity is conducive to improving the growth performance of laying hens.

Table 3 OK but 20 wk Body slop length P value P<0.050 should be checked, there is a significant difference, there is no lettering.

The P value in “P=0.050”, and “P < 0.05” was set as the level of significance in this article. So there was no letter.

Table 4 OK

CV values ​​are given in Figures 3 and 4 but what is the importance of the CV value is not mentioned in the article. Since the necessary parameter should be explained, if not, it should be removed.

-Be revised.

We believe that it is an important indicator and related explanations are supplemented in the discussion section.

Table 5 OK

Figure 5 and 6 OK

Why is the CV value for egg weights important in Table 6? What does it mean? Is it necessary for the article?

The coefficient of variation in egg weight reflects the uniformity of egg weight in the layer population, and thus reflects the uniformity of the group. In this experiment, individual data were recorded for phenotypic traits of laying hens, such as body weight, egg weight, laying characteristics, egg production characteristics and so on, in order to analyze the individual differences of laying hens with different BW at 6 wks of age. So we believe that it is an important indicator.

Table 7 OK.

In line 256. Columns with different superscripts (a, b) are significantly different at P < 0.05 should be added above figure

-Done as requested.

There is no statement in the discussion about the CV of egg counts, but in the conclusion section it is stated with a sentence that the CV of egg counts was worse in the LWG group. What should readers of this article understand about the CV value or what it means should be stated in more detail in the discussion.

-Be revised.

The CV of individual egg number of reflects the uniformity of the production performance, a smaller CV indicates that the laying hens lay more evenly and is expected to have a better egg production.

We believe that it is an important indicator and related explanations are supplemented in the discussion section.

Reviewer 5 Report

Comments and Suggestions for Authors

Title

  1. The title suffers from two critical issues: Overlapping prepositions ('upon...on...regarding') introduce ambiguity in variable relationships; Weak focus due to conflated temporal and causal elements.

Please restructure the title to make it clear and concise.

Abstract

  1. Line 27  BW should be written in full with its abbreviation when first mentioned.
  2. Line 32-59 The Results section is too verbose. Please present the key findings in a logical sequence.

Introduction

  1. BW should be written in full with its abbreviation when first mentioned.
  2. A clear hypothesis must be explicitly stated in the introduction.
  3. Please supplement the theoretical basis supporting the hypothesis that body weight at brooding period completion could serve as a more reliable indicator of overall production performance in laying hens, including potential physiological mechanisms and relevant literature evidence

Materials and Methods

  1. Line 118 Please provide the dietary formulations (composition and nutrient levels) for weeks 1-5 of the trial period.
  2. Line143 Please clarify the scientific rationale for selecting only 15,20 weeks of age for body size measurements in comparison.

Results

  1. Please present the specific BW and CV data in Table/Figure format with numerical values for Result 3.1
  2. Line 256  Check.

Discussion

  1. Line300-341,The first paragraph is too lengthy with unclear logic and should be restructured for better clarity and focus.The remaining paragraphs also suffer from similar issues.

The Discussion section should focus on interpreting how body weight at the end of the brooding period serves as a more comprehensive indicator of overall laying performance, with direct reference to both your experimental results and supporting literature evidence.

Conclusions

  1. The Conclusion section should be concise and focused, briefly summarizing the key findings regarding the relationship between body weight at brooding period completion and overall production performance in laying hens

References

13 Check style as journal require. Lin477

Author Response

Comments and Suggestions for Authors

Title

1.The title suffers from two critical issues: Overlapping prepositions ('upon...on...regarding') introduce ambiguity in variable relationships; Weak focus due to conflated temporal and causal elements.

Please restructure the title to make it clear and concise.

-Be revised.

Influence of body weight at the end of the brooding period on the productive performance in Hyline Brown laying hens from 6 to 72 weeks of age

Abstract

  1. Line 27  BW should be written in full with its abbreviation when first mentioned.

-Done as requested.

We have checked and revised throughout the manuscript.

  1. Line 32-59 The Results section is too verbose. Please present the key findings in a logical sequence.

-Be revised.

The heavier BW in the normal weight group (NWG) at 6 wks of age compared to lighter weight group (LWG) birds continued until 22 wks (P < 0.05). A smaller coefficient of variation (CV) for BW of chicks in the LWG was detected at 19 (P < 0.01), 20 (P < 0.01), and 21 (P < 0.05) wks of age. The body slope length and the shank circumference of pullets in the LWG were smaller than in the NWG at the age of 15 wks (P < 0.01), but the difference gradually disappeared by 20 wks of age (corresponding P = 0.050 and 0.916). The LWG presented raised ages of hens when producing the first egg and 5% eggs (P < 0.01), while the CV for the age at first egg decreased, compared with the CV in the NWG (P < 0.05). The total egg number (P < 0.05), laying rate (P < 0.05), and egg mass (P < 0.01)of laying hens in the LWG decreased at the age of 18-72 wks, and the CV for individual egg numbers (P < 0.05) increased compared with the CV in the NWG. Compared with the normal weight hens, the proportion of lighter weight hens laying more than 250 eggs at the age of 18-72 wks was significantly reduced (P < 0.05, 69.52% vs. 87.38%), while the proportion of hens laying less than 200 eggs was significantly increased (P < 0.05, 24.97% vs. 3.76%).

Introduction

  1. BW should be written in full with its abbreviation when first mentioned.

-Done as requested.

We have checked and revised throughout the manuscript.

  1. A clear hypothesis must be explicitly stated in the introduction.

-Done as requested.

The hypothesis tested in this study was that BW at the end of brooding period may serve as an important phenotypic indicator for evaluating laying performance and early elimination of unqualified laying hens in layer production.

  1. Please supplement the theoretical basis supporting the hypothesis that body weight at brooding period completion could serve as a more reliable indicator of overall production performance in laying hens, including potential physiological mechanisms and relevant literature evidence

-Done as requested. Related explanations are supplemented in the paper.

As for laying pullets, the growth, development and BW in the brooding, rearing and pre-laying stages serve as pivotal influencing factors for both physical and sexual maturity when the egg-laying cycle starts, potentially directly impacting on the overall laying performance. Therefore, many studies on evaluating BW of laying pullets near the egg-laying onset for its effects on egg quality, sexual maturity, and the whole laying cycle production performance have been conducted. However, most studies have found that compared to hens with lighter BW at the onset of lay, heavier weight hens have higher average egg weight (EW) and cumulative egg mass as well as better eggshell quality, but the laying rate in the whole laying cycle did not differ.

Materials and Methods

  1. Line 118 Please provide the dietary formulations (composition and nutrient levels) for weeks 1-5 of the trial period.

Not provided.

All laying pullets from 1 to 5 wks of age were fed the same diet and the nutritional levels of the diet were not determined. The trial commenced at the end of the 6th week, so we did not provide the diet formulations and nutritional levels for 1 to 5 wks of age.

  1. Line143 Please clarify the scientific rationale for selecting only 15,20 weeks of age for body size measurements in comparison.

According to the Hyaline Brown breeding manual, 15 weeks of age is the end point of the growing period, and 20 weeks of age is the day of 50% laying rate. In the previous research, our team found that the pre-laying and early laying periods are important stages for the growth and development of small-sized laying hens, preparing for egg production. Therefore, we chose to measure the body size of laying pullets at 15 and 20 weeks of age.

Results

  1. Please present the specific BW and CV data in Table/Figure format with numerical values for Result 3.1

We have organized the data of BW and BW CV into a table format, but it didn't feel as reflective of the overall trend as a picture format. Therefore, we did not replace it in the article, and we will do as required if the editor requires it to be replaced with a table format.

Table 3. The BW (g) and BW CV (%) of hens with different BWs on completion of the brooding phase.

Items

Normal weight hens

Lighter weight hens

P-value

BW (g)

6 wk

375.8±4.42a

295.0±2.13b

<0.001

12 wk

1105.0±5.23a

1000.1±6.24b

<0.001

15 wk

1368.7±8.35a

1272.9±19.03b

0.001

16 wk

1413.6±6.15a

1339.9±13.70b

0.011

17 wk

1474.5±10.38a

1385.2±12.20b

0.007

18 wk

1520.6±16.43a

1453.0±17.95b

0.044

19 wk

1595.5±12.56a

1504.8±16.36b

0.002

20 wk

1693.2±19.25a

1591.5±23.26b

0.010

21 wk

1761.3±17.33a

1650.9±23.44b

0.005

22 wk

1827.6±12.42a

1763.2±20.31b

0.020

23 wk

1888.0±11.57

1854.6±25.31

0.213

24 wk

1908.2±13.99

1908.6±12.80

0.986

27 wk

1996.9±9.69

2004.5±27.66

0.769

BW CV (%)

6 wk

10.8±0.72a

7.7±0.38b

0.011

12 wk

7.2±0.61

6.8±0.58

0.685

15 wk

6.5±0.40

6.6±1.58

0.990

16 wk

7.3±0.47

5.7±0.70

0.145

17 wk

6.5±0.10

5.8±0.50

0.388

18 wk

7.2±0.33

6.5±0.52

0.346

19 wk

7.6±0.21a

6.2±0.43b

0.009

20 wk

8.5±0.24a

6.1±0.40b

0.001

21 wk

8.7±0.32a

6.8±0.63b

0.014

22 wk

7.7±0.30

6.6±0.91

0.316

23 wk

6.6±0.32

6.5±1.02

0.879

24 wk

6.3±0.33

6.2±0.55

0.947

27 wk

7.3±0.78

6.7±1.17

0.694

  1. Line 256  Check.

-Be revised.

"Columns with different superscripts (a, b) are significantly different at P < 0.05."

Discussion

  1. Line300-341, the first paragraph is too lengthy with unclear logic and should be restructured for better clarity and focus. The remaining paragraphs also suffer from similar issues.

The Discussion section should focus on interpreting how body weight at the end of the brooding period serves as a more comprehensive indicator of overall laying performance, with direct reference to both your experimental results and supporting literature evidence.

-Done as requested.

Conclusions

  1. The Conclusion section should be concise and focused, briefly summarizing the key findings regarding the relationship between body weight at brooding period completion and overall production performance in laying hens

-Done as requested.

References

  1. Check style as journal require. Lin477

-Be revised.

  1. Zhao, X. Y.; Zhang, Y.; He, W.; Wei, Y. H.; Han, S. S.; Xia, L.; Tan, B.; Yu, J.; Kang, H. Y.; Ma, M. G.; Zhu, Q.; Yin, H. D.; Cui, C. Effects of small peptide supplementation on growth performance, intestinal barrier of laying hens during the brooding and growing periods. Front. Immunol. 2022, 13, 1–14.

Round 2

Reviewer 3 Report

Comments and Suggestions for Authors

Dear Authors,

Thanks for your response. I have still similar ideas about the originality of the study. However your paper has a large data set related about production period of Hyline Brown. I only think your paper could potetially contribute to the other researcher ideas'. Therefore, I recomment to publish it.

Author Response

Comments and Suggestions for Authors

Thanks for your response. I have still similar ideas about the originality of the study. However your paper has a large data set related about production period of Hyline Brown. I only think your paper could potetially contribute to the other researcher ideas'. Therefore, I recomment to publish it.

Dear professor,

Thanks for your approval.

Submission of a paper is also a learning process for the authors. We can learn the opinions of reviewers to improve ourselves. We will further study the mechanism of the performance difference between the two groups or possible solutions to minimize these negative states.

Thanks!

Reviewer 4 Report

Comments and Suggestions for Authors

The manuscript was evaluated and it was determined that the necessary corrections were made. P values ​​should be shown in lower case letters in the article. P= should not be in lines 33 and 216, P< should be

Author Response

Comments and Suggestions for Authors

The manuscript was evaluated and it was determined that the necessary corrections were made.

P values ​​should be shown in lower case letters in the article. P= should not be in lines 33 and 216, P< should be

Dear professor,

Thanks for your approval.

-Done as requested.

We have checked and revised throughout the manuscript. We have highlighted the changes in the revised manuscript.

Reviewer 5 Report

Comments and Suggestions for Authors

Can be accepted.

Author Response

Comments and Suggestions for Authors

Can be accepted.

Dear professor,

Thanks for your approval.